# Challenges in the Differential Diagnosis of COVID-19 Pneumonia: A Pictorial Review

**DOI:** 10.3390/diagnostics12112823

**Published:** 2022-11-16

**Authors:** Cristina Maria Marginean, Mihaela Popescu, Corina Maria Vasile, Ramona Cioboata, Paul Mitrut, Iulian Alin Silviu Popescu, Viorel Biciusca, Anca Oana Docea, Radu Mitrut, Iulia Cristina Marginean, George Alexandru Iacob, Daniela Neagoe

**Affiliations:** 1Department of Internal Medicine, University of Medicine and Pharmacy of Craiova, 200349 Craiova, Romania; 2Department of Endocrinology, University of Medicine and Pharmacy of Craiova, 200349 Craiova, Romania; 3Department of Pediatric and Adult Congenital Cardiology, Bordeaux University Hospital, 33600 Pessac, France; 4Department of Pneumology, University of Pharmacy and Medicine of Craiova, 200349 Craiova, Romania; 5Department of Toxicology, University of Medicine and Pharmacy of Craiova, 200349 Craiova, Romania; 6Department of Cardiology, University and Emergency Hospital, 050098 Bucharest, Romania; 7Faculty of Medicine, University of Medicine and Pharmacy of Craiova, 200349 Craiova, Romania

**Keywords:** COVID-19, pneumonia, differential diagnosis, differential diagnosis

## Abstract

COVID-19 pneumonia represents a maximum medical challenge due to the virus’s high contagiousness, morbidity, and mortality and the still limited possibilities of the health systems. The literature has primarily focused on the diagnosis, clinical-radiological aspects of COVID-19 pneumonia, and the most common possible differential diagnoses. Still, few studies have investigated the rare differential diagnoses of COVID-19 pneumonia or its overlap with other pre-existing lung pathologies. This article presents the main radiological features of COVID-19 pneumonia and the most common alternative diagnoses to establish the vital radiological criteria for a differential diagnosis between COVID-19 pneumonia and other lung pathologies with similar imaging appearance. The differential diagnosis of COVID-19 pneumonia is challenging because there may be standard radiologic features such as ground-glass opacities, crazy paving patterns, and consolidations. A multidisciplinary approach is crucial to define a correct final diagnosis, as an overlap of COVID-19 pneumonia with pre-existing lung diseases is often possible and suggests possible differential diagnoses. An optimal evaluation of HRTC can help limit the clinical evolution of the disease, promote therapy for patients and ensure an efficient allocation of human and economic resources.

## 1. Introduction

The new coronavirus (COVID-19) has caused a severe pandemic involving several nations. It is a significant concern and an enormous challenge, not only for doctors and researchers but also for the whole population.

It is caused by SARS-CoV-2, an RNA virus of the Sarbecovirus family, similar to the SARS virus [1]. This virus binds to the human angiotensin-converting enzyme receptor (hACE2), triggering general and respiratory symptoms [2]. The primary transmission mode is human-to-human, and the average incubation period is about four days [3]. Symptoms most observed in different cohorts of COVID-19 patients are fever (83–98%), followed by fatigue (70%) and dry cough (59%) [4]; gastrointestinal symptoms were relatively rare.

The disease is usually mild (80%), and the recovery period is, on average, about two weeks [5]. COVID-19 frequently occurs in elderly and middle-aged men, with the highest incidence of death (8–15%) amongst people aged >80 years. The onset of illness begins with fever, dry cough, fatigue, and myalgia, progressing to dyspnea and ARDS within 6 and 8 days of exposure, respectively [2,6]. The underlying comorbidities also increase the mortality rates of COVID-19. The poor prognosis includes age, comorbidities, severe lymphopenia, high CRP, and dimer D > 1 μg/L [7]. The overall mortality rate varies between 1.5 and 3.6% [8].

Regardless of the constant and vigorous efforts of researchers and the medical community, we are still far from eradicating, curing, or preventing by vaccination. Therefore, education, strict prevention, and control measures are the only weapon available to combat the extent of this ongoing infectious disease.

## 2. Pathophysiology

The infection progresses through a replication stage in the first few days, followed by an adaptive immunity phase in the next few days [9]. In the replication stage, the virus replicates, leading to a flu-like illness characterized by mild symptoms due to the direct cytopathic effect of the virus on type II alveolar cells. In the adaptive immune stage, viral levels decrease as the immune system becomes activated. However, the cytokine storm leads to tissue destruction and clinical damage—which explains why patients remain relatively well in the early stages before deteriorating suddenly. The implications of this are early initiation of antiviral therapies for better outcomes and the use of immunosuppressive therapies in targeting adaptive immunity [4].

## 3. Incubation Period (IP)

The incubation period for COVID-19 has been defined as the potential interval between the first contact with the source of transmission (wildlife or suspected or confirmed case) and the first possible appearance of symptoms (cough, fever, fatigue, or myalgia). Generally, this is less than 14 days after exposure, with an average incubation period of four days (2–7 days)—50% of cases are reported within this period [10].

## 4. Demographic Features and General Symptoms of COVID-19

The spectrum of disease ranges from mild disease (mild pneumonia) in 81% of cases, with a typical recovery period of about two weeks, to severe illness (dyspnea, hypoxia) with >50% pulmonary affection on imaging in 14% of patients, with a recovery period of approximately 3–6 weeks, until critical illness (ARDS, sepsis, septic shock, or MODS) in 5% of cases; this was observed in data from 44,500 confirmed cases of COVID-19 [5].

COVID-19 mainly affects men (58.1%), and the predominantly affected age groups are middle-aged and elderly, with fewer cases reported in children (0.9–2%). The elderly are more seriously affected and have a higher mortality rate (8–15%) [11].

Most studies on hospitalized patients reveal that the common symptoms of COVID-19 are fever (83–98%), fatigue (70%), dry cough (59%), anorexia (40%), myalgia (35%), dyspnea (31%) or a productive cough (27%) [4,11,12,13]. Fever in COVID-19 has been described in different cohorts, with variations between low fever (37–38 °C) and persistent fever lasting up to 14 days. In a study of 1099 patients, only 43.8% had a fever on admission and 88.7% during hospitalization [13].

Gastrointestinal symptoms such as nausea, vomiting (5%), and diarrhea (3.8%) are less common, differentiating COVID-19 from SARS and MERS. Asymptomatic infection was also observed, but the frequency is unknown.

## 5. Clinical Evolution

Symptoms of COVID-19 (Figure 1) initially begin with fatigue, intermittent or prolonged fever, myalgia, dry cough, and shortness of breath, symptoms improved with early administration of conservative therapy or worsen and progress to dyspnea and productive cough [4]. The average time to onset of dyspnea in various studies was 6 days after exposure.

The median time to admission, development of ARDS, and need for mechanical ventilation and ICU care was 8, 8.2, and 10 days, respectively. The average duration of hospitalization from the onset of the disease was 22 days (18–25 days), while the average time until death was 18 days (15–22 days). The median duration of fever was 12 days (8–13 days) and cough persisted for 19 days (12–23 days) in survivors [11].

Complications like bilateral pneumonia, ARDS, septic shock, acute cardiac injury, acute kidney injury, and secondary infections occurred in 12–19 days after onset. The medium duration of viral clearance was 20 days (17–24 days) from disease onset in survivors vs. non-survivors, who continued to eliminate the virus until exitus [4,11].

## 6. Underlying Comorbidities Related to the Evolution of COVID-19

The main comorbidities that are worsening the evolution of COVID-19, with increased disease severity, use of mechanical ventilation, and increased mortality, include uncontrolled hypertension, diabetes, coronary heart disease, cerebrovascular disease, chronic obstructive pulmonary disease, chronic liver disease, and other such as cancer, chronic kidney disease, and immunodeficiency. Guan et al. (2020) [13] revealed in a study that hypertension (15%) was predominant, also diabetes (7.4%), probably due to the hACE-2 receptor polymorphism in the Asian population (Figure 2).

## 7. Laboratory Findings

Significant laboratory findings (Figure 3), as in all other viral respiratory diseases, include lymphopenia, elevated liver enzymes, and elevated D dimers [4].

Severe lymphopenia, C-reactive protein (CRP), D dimers (>1 μg/L), IL-6, ALT, serum ferritin, lactate dehydrogenase, creatine kinase, troponin are markers with high sensitivity, creatinine, prothrombin time, and procalcitonin being associated with higher mortality [13].

## 8. Prognostic Factors in COVID-19

The strong independent predictors of elevated mortality are age (over 70 years), comorbidities such as uncontrolled hypertension, diabetes, coronary heart disease, chronic obstructive pulmonary disease, neoplastic diseases, severe lymphopenia, and elevated D-dimer [7]. Additional negative prognostic factors are elevated C-reactive protein, LDH, ALT, serum ferritin, IL-6, and high-sensitivity cardiac troponin [14,15]. Serum sodium, glomerular filtration rate, and creatinine are useful in predicting the clinical outcome of patients with moderate forms of the disease [16].

## 9. Diagnostic Imaging

Chest radiographs can be used to diagnose COVID-19 due to their availability and low cost. It is helpful in the emergency room and in cases where the patient cannot be mobilized for computed tomography. At the onset, chest X-ray has low sensitivity and may even be normal [17]. Ground-glass opacities may not be thick enough to be evident by radiography, and, if they have a basal and retro-cardiac location, they may be hidden by the diaphragm or mediastinal structures [18].

Computed tomography is a method with higher sensitivity and specificity compared to chest X-ray, and it can also discover abnormalities during the early stages of COVID-19. It is routinely used in patients with clinical suspicion of COVID-19 in the screening and differential diagnosis of pneumonia. Still, a normal chest CT scan does not exclude a positive diagnosis of COVID-19.

High-resolution computed tomography (HRCT) reveals single or multiple GGOs with a predominantly subpleural distribution, crazy pavement, or segmental pulmonary consolidations [19].

Ultrasound (US) can be used in the triage of suspected patients to assess the severity and evolution of the disease, being a method without the risks of radiation, especially in children and pregnant women, but having disadvantages for the examiner (increased exposure time). US is useful in evaluating consolidations and interstitial lung involvement, with increased sensitivity and specificity than chest radiographs, as well as in the differential diagnosis of cardiogenic versus noncardiogenic acute pulmonary edema [20].

Although a susceptible method in detecting COVID-19 pneumonia, the US features are not pathognomonic, being similar in all interstitial and alveolar lung diseases, including viral pneumonia, idiopathic pulmonary fibrosis, hypersensitivity pneumonia, heart failure, and diffuse alveolar hemorrhages. Characteristics for COVID-19 US features are pleural thickening; vertical B lines determined by the decrease in lung aeration secondary to interstitial lung damage (focal, multifocal, or confluent); consolidations with aeric bronchograms, pleural effusions (their presence may lead to differential diagnoses such as bacterial pneumonia or congestive heart failure) [20].

Magnetic resonance imaging (MRI) is used to diagnose cardiac or central nervous system complications of COVID-19. On MRI, lung parenchymal changes appear as regions of increased signal intensity, corresponding to ground glass opacities, or consolidations also highlighted by chest X-ray or CT [21].

### COVID-19 Pneumonia

COVID-19 pneumonia has been divided into four stages, which may have overlapping radiological elements [22]:

1. Early phase/Stage 1—days 0–4. Ground glass opacities represent the main radiological characteristic [12] (Figure 4A);

2. The progressive phase/Stage 2 refers to days 5–8, and the hallmark is a cobblestone appearance (Figure 4B) coexisting with extensive ground-glass opacities and condensation foci [23];

3. Peak phase/Stage 3 is typical for days 9–13, and CT shows pulmonary condensations (Figure 4C), sometimes surrounded by a halo of ground glass.

4. The absorption phase/Stage 4 begins around day 14; areas of ground glass together with linear condensations are appreciable (Figure 4D).

Opacities are usually bilateral and subpleural, having an apicobasal gradient of distribution. Additional radiological features are enlargement of the peripheral pulmonary vessels, while pleural effusions, pulmonary nodules, and mediastinal lymphadenopathy are rare [24].

Clinical conditions may worsen suddenly, with patients presenting wheezing, dyspnea, and tachypnea with low blood oxygen saturation. These features indicate the progression of COVID-19 pneumonia to ARDS (acute respiratory distress syndrome). ARDS is highlighted on HRCT as patchy ground-glass confluent areas and pulmonary condensations [19]. Clinical and radiological monitoring is key to the early identification and treatment of ARDS in COVID-19 pneumonia [23,24]. Opacities are usually bilateral and subpleural, with an apicobasal gradient of distribution. Additional radiological features are enlargement of the peripheral pulmonary vessels, while pleural effusions, pulmonary nodules, and mediastinal lymphadenopathy are rare [24,25].

## 10. Differential Diagnosis of COVID-19 Pneumonia

Prompt recognition, isolation, and rapid treatment initiated in suspected cases of COVID-19 are essential during this pandemic. Failure to recognize alternative differential diagnoses and co-infections (given the similarity of symptoms and imaging with other systemic conditions) can lead to delays in diagnosis and treatment [26].

Establishing the diagnosis of COVID-19 on clinical and radiological criteria alone may be an incomplete diagnostic strategy; one patient in five with symptoms of respiratory tract infection and altered chest CT will be diagnosed with an alternative condition than COVID-19, such as other diseases infectious and non-infectious [27]. RT-PCR negativity for SARS-CoV-2 in nasopharyngeal specimens should follow prompt further investigation of the presence of the virus in different models, such as induced sputum, feces, and serum while searching for likely differential diagnoses [28].

COVID-19 must be differentiated from other acute respiratory diseases such as viral pneumonia, respiratory infections caused by influenza or parainfluenza viruses, respiratory syncytial viruses, rhinovirus, adenoviruses but also bacterial pneumonia [29].

Other possible differential diagnoses would be acute heart failure [30,31], pulmonary embolism and exacerbation of COPD, and idiopathic interstitial pneumonia [32].

### 10.1. Bacterial Pneumonia

Radiological differential diagnosis: single consolidation with air bronchogram, usually presenting as lobar pneumonia (typical pneumonia: *Staphylococcus aureus*, *Streptococcus pneumoniae*, *Moraxella catarrhalis*, *Enterobacteriaceae*) [33,34,35] (Figure 5A); multifocal pneumonia presents ground glass opacities and consolidations; usually coexists with centrolobular nodules and bronchovascular bundle thickening (atypical pneumonia: *Mycoplasma pneumoniae*, *Chlamydia pneumoniae*) [34,36] (Figure 5B).

*Mycoplasma pneumoniae* can cause multiple bilateral ground-glass opacities like COVID-19 but occurs mainly in children, while COVID-19 pneumonia is less common and severe in the pediatric field.

Additional findings are centrolobular nodules, cavities, pneumatoceles (more common in *S. aureus* pneumonia); hilomediastinal lymphadenomegaly; pleural effusions [35].

### 10.2. Viral Pneumonia

Viral pneumonia is defined as a diverse entity, and it is mainly the current epidemic context that suggests the origin of COVID-19. The treatment proved to be similar at present [37]. Radiological characteristics in differential diagnosis: preferential central-parenchymal involvement (Influenza type A, Adenovirus, Hantavirus) (Figure 6A); additional findings, such as centrolobular nodules and bronchial wall thickening (RSV, MERS, Influenza type A); coexisting pulmonary edema (Hantavirus); pleural effusions (RSV, MERS); hilomediastinal lymphadenomegaly (Influenzae type A) [38,39,40].

### 10.3. Pneumocystis Pneumonia

Pneumocystis pneumonia is an opportunistic fungal infection that mainly affects immunodeficient patients affected by AIDS or undergoing immunosuppressive therapy (Figure 6B) [41]. History and laboratory tests are helpful but often insufficient for a differential diagnosis of COVID-19 pneumonia [37,42,43].

Imagistic diagnosis characteristics are symmetrical, centroparenchymal and perihilar, ground-glass, confluent opacities, generally with subpleural reduction, a predilection for the upper lobes, sometimes cobblestone appearance; pneumatoceles [24,44].

### 10.4. Aspergillosis

It is caused by *Aspergillus Fumigatus* and generally affects immunocompromised patients with severe neutropenia. Radiological diagnosis highlights: ground glass opacities and cobblestone aspect are not typical and do not precede condensations, which frequently show a surrounding halo of ground glass (halo sign) [45] (Figure 6C).

In case of consolidations without ground glass halo and the absence of other ground glass opacities, COVID-19 pneumonia is unlikely; lymphadenopathy and pleural effusions are also present.

### 10.5. Cardiovascular Pathologies

#### 10.5.1. Acute Pulmonary Edema

There are two pathophysiological and radiological phases recognized in the development of cardiogenic pulmonary edema: interstitial and alveolar.

In addition to anamnesis, radiological features (Figure 7A,B) for differential diagnosis are the possible coexistence of ground-glass opacities, pavement, and consolidations with different times of occurrence about COVID-19 pneumonia [30]; the pattern of crazy pavement—diffuse, bilateral, with central disposition—and ground-glass opacities with subpleural preservation. Condensations are late and generally coexist with pleural effusions; bilateral pleural effusions, more evident in the alveolar phase of edema [46]; mediastinal lymphadenopathy; cardiomegaly.

Cases of acute myocarditis related to COVID-19 infection have been reported. The standard features of COVID-19 pneumonia and pulmonary edema should always raise the suspicion of myocarditis, especially in young patients.

#### 10.5.2. Differential Diagnosis of COVID-19 Pneumonia with Acute Heart Failure

COVID-19 pneumonia usually symptoms are fever, cough, and dyspnea. Although typical imaging features for COVID-19 pneumonia are specified on chest CT (ground-glass appearance), these features can sometimes be confused with other conditions [31].

One of these clinical entities is acute heart failure, characterized by fluid accumulation in the interstitial and alveolar spaces due to increased hydrostatic pressure in the pulmonary vessels. Acute pulmonary edema caused by heart failure can mimic several diseases on chest CT, leading to delays in diagnosing and treating these patients [47].

Anamnesis, history of exposure to SARS-CoV-2, and symptoms such as fever, cough, and fatigue were reported in a significantly higher number of patients with COVID-19 pneumonia than in those with acute heart failure.

The distribution of lesions on chest x-ray and CT is different, with the predominance of central lesions in the AHF compared to patients diagnosed with COVID-19 pneumonia [31,46,47].

Pleural effusion and cardiomegaly, mediastinal lymphadenopathy, septal thickening, and dilation of pulmonary veins are found in a significantly higher percentage of patients with AHF.

The distribution of lesions on chest CT, pleural effusion, and cardiomegaly can provide important information to clinicians in evaluating the differential diagnosis [31,46,47].

Biological tests reveal it is significantly lymphopenia. In addition, patients with COVID-19 pneumonia had significantly increased levels of CRP, ferritin, LDH, and CK compared to patients with AHF, and the level of NT—proBNP is considerably higher in the group of patients with AHF. Compared to AFH, patients with COVID-19 pneumonia have higher LDH, CK, ferritin, and CRP [48].

### 10.6. Vasculitis

ANCA-associated small-vessel vasculitis frequently occurs with predominant pulmonary involvement and may cause diffuse alveolar hemorrhage (DAH) (Figure 8A,B).

History and laboratory tests may help confirm clinical suspicion. Imagistic features are: DAH, appearing as ground glass opacities or condensation foci if the bleeding is massive, is more prominent in the perihilar region and the lower lobes [48]; opacities or symmetrical, bilateral ground glass consolidations, migratory or transient; coexistence of bronchial and tracheal thickening in granulomatosis with polyangiitis [49]; coexistence of bronchial and bronchiolar thickening or centrolobular nodules in eosinophilic granulomatosis with polyangiitis; pulmonary edema, secondary to heart damage in eosinophilic granulomatosis with polyangiitis [39,48,49]; pleural effusions.

### 10.7. Hypersensitivity Pneumonia

It is an interstitial pathology caused by the inhalation and repeated sensitization of a wide range of inorganic and organic antigens, occupationally or from the environment. Radiological characteristics for differential diagnosis (Figure 9A): centroparenchymal and centrolobular ground glass opacities; the rare appearance of cobblestone and pulmonary condensations; coexistence of other radiological aspects such as oligemia, cysts, centrolobular emphysema, and centrolobular micronodules; coexistence of ground glass opacities, preserved lung regions and air trapping on HRCT (Figure 9B); centrolobular fibrosis, architectural distortions, traction, and bronchiectasis in the chronic phase; mediastinal lymphadenopathy.

### 10.8. Eosinophilic Pneumonia

Eosinophilic pneumonia represents a distinct group of pulmonary diseases accompanied by peripheral eosinophilia. Laboratory tests and history are crucial for a correct differential diagnosis. HRTC is essential for positive and differential diagnosis [1].

Imaging characteristics for differential diagnosis are (Figure 10A,B): the absence of the cobblestone in simple pulmonary eosinophilia (SPE), also known as Loeffler syndrome); pleural effusions, centrilobular nodules, and thickening of lar bundles in acute eosinophilic pneumonia [50]; centrolobular consolidations with occasional frosted glass and cobblestone opacities in chronic eosinophilic pneumonia; additional findings in chronic eosinophilic pneumonia are represented by nodules, atelectasis, band opacities, and pleural effusions [51,52].

### 10.9. Aspiration Pneumonia

Aspiration pneumonia is determined by the aspiration of substances inside the airways and pulmonary parenchyma. Radiological findings may differ, but the history is often sufficient for diagnosis [53].

#### 10.9.1. Pneumonia with Fluid Aspiration

Frequently, these patients are dysphagic, and their meals are liquid. Characteristic images or the differential diagnosis are ground glass opacities; in the late stages, fibrotic architectural distortions [54].

#### 10.9.2. Lipoid Pneumonia

Lipoid pneumonia is acute or chronic reactive pneumonia resulting from endogenous lipid accumulation or exogenous lipid aspiration. Chronic lipoid pneumonia requires a differential diagnosis with COVID-19, while history is generally sufficient for a precise diagnosis of acute pneumonia [55].

A history of lipid inhalation is necessary to diagnose lipoid pneumonia, and a comparison with previous chest examinations is essential.

Imaging diagnosis is necessary (Figure 11A,B): ground glass opacities and condensation predominantly in the middle as well as in the lower lobes [56]; condensations usually show very low CT attenuation concerning their fat content; fibrosis in the chronic stage [57,58].

### 10.10. Pulmonary Alveolar Proteinosis

Pulmonary alveolar proteinosis-PAP is a syndrome caused by an accumulation of surfactant in the pulmonary alveoli. It can be primary in most cases or secondary to toxic inhalation syndromes, hematological neoplasms, and immune deficiency. History, laboratory tests, and comparison with previous HRCT are helpful [59,60].

CT differential diagnosis is highlighted by (Figure 12A,B): mainly centro-parenchymal and perihilar cobblestone areas; the juxtaposition of severely affected secondary lobules and normal secondary lobules [61,62]; rarely, condensations with air bronchogram in severe forms; progressive fibrotic changes; pleural effusions, cardiomegaly, and lymphadenopathy, which are characteristics of complicated PAP [60].

### 10.11. Drug-Induced Pulmonary Pathology

The history of drug use is crucial to diagnosing and establishing appropriate treatment.

Among the most common drugs/drugs causing associated lung diseases are amiodarone and methotrexate (pneumonia in the organization); immunosuppressants (hypersensitivity pneumonia); heroin (eosinophilic pneumonia, pulmonary hemorrhage, pulmonary edema); cocaine (pulmonary edema) [63].

### 10.12. Differential Diagnosis between COVID-19 Pneumonia and Idiopathic Interstitial Pneumonia (IIP)

COVID-19 pneumonia and IIP represent complex pulmonary pathologies, and the differential diagnosis is frequently challenging.

The radiological and tomographic aspects must always be correlated with the anamnesis, clinical, and laboratory data.

Idiopathic interstitial types of pneumonia and COVID-19 pneumonia are different entities but share some similar radiological features.

The association between typical symptoms and correlated radiological images can suggest the diagnosis of COVID-19 and warrant the isolation of patients to avoid the spread of infection [64].

Yet, a discordance between the patients’ anamnesis and the imaging should suggest a possible differential diagnosis [64,65]. Comparison with previous CT scans is crucial to identify possible chronic or long-standing radiological findings of IIP. In general, the presence of typical and/or additional radiological changes that are less frequent or rare in COVID-19 pneumonia is found in IIP (Figure 13A,B) [65,66]: the migration of condensation foci (organizing pneumonia); preferential involvement of the lobular periphery, resulting in a “peri-lobular pattern” (pneumonia in organization); ground glass opacities during disease exacerbation; relative reduction of the subpleural lung tissue (nonspecific interstitial pneumonitis, organizing pneumonitis); predominance in the upper fields (bronchiolitis associated with interstitial lung disease, lymphoid interstitial pneumonia, pleuropulmonary fibroelastosis) [67]; apicobasal gradient and heterogeneous lung involvement (idiopathic pulmonary fibrosis); clear demarcation between the healthy lung parenchyma and the affected parenchyma (idiopathic pulmonary fibrosis) [68]; the coexistence of other radiological findings such as centrolobular nodules (bronchiolitis associated with interstitial lung disease, organizing pneumonia) and thin-walled cysts (interstitial lymphoid pneumonia) [69]; the presence of fibrosis that can be appreciated as parenchymal distortion, bronchial traction, and/or honeycombing (idiopathic pulmonary fibrosis, nonspecific interstitial pneumonitis); pleural thickening (pleuropulmonary fibroelastosis); pleural effusions (exudative acute interstitial pneumonia, organizing pneumonia).

The differential diagnosis of COVID-19 pneumonia with IIP is challenging because these entities may share common radiological aspects. Therefore, a multidisciplinary approach is crucial to arrive at a final and correct diagnosis [70,71].

### 10.13. Acute Pulmonary Embolism (APE)

APE, frequently secondary to lower limb thrombosis, has an acute clinical onset and can cause pulmonary infarction (Figure 14A,B).

An adequate history and radiological examinations such as Angio CT are crucial to identify luminal defects of the pulmonary vessels.

Radiological diagnosis includes [25]: different phases of infarct maturation in correlation with onset time; the segmental form of infarcts is in the vascular territories of the affected vessels; the presence of embolism, vessel occlusion, and residual peripheral clot deposition. Extensive pulmonary thromboembolism in severe cases of COVID-19 pneumonia, mediated by the endothelial tropism of COVID-19, has been suspected [72]. Fortunately, heparin, the first therapy for APE, has been shown to work in patients affected by severe forms of COVID-19 pneumonia [73,74].

COVID-19 pneumonia and pulmonary embolism can coexist; in particular, a person could have symptoms of acute pulmonary embolism but could also be infected with COVID-19, with typical CT-appearing ground-glass opacities being the only sign in this patient [75].

## 11. Conclusions

Early detection of the condition is essential for optimal treatment implementation, patient isolation, and effective public health surveillance.

Along with clinical symptoms and laboratory data, lung imaging plays a crucial role in identifying the radiological features of COVID-19 pneumonia and assessing possible differential diagnoses and overlap with pre-existing chronic lung disease. Imaging methods, such as HRCT, provide comprehensive and complete information and must be integrated into the clinical context. More studies regarding the characteristics of lung lesions produced by COVID-19 on other causative factors are vital to identify distinctive features that may support a positive diagnosis and exclude the varied differential diagnosis of COVID-19.

## Figures and Tables

**Figure 1 diagnostics-12-02823-f001:**
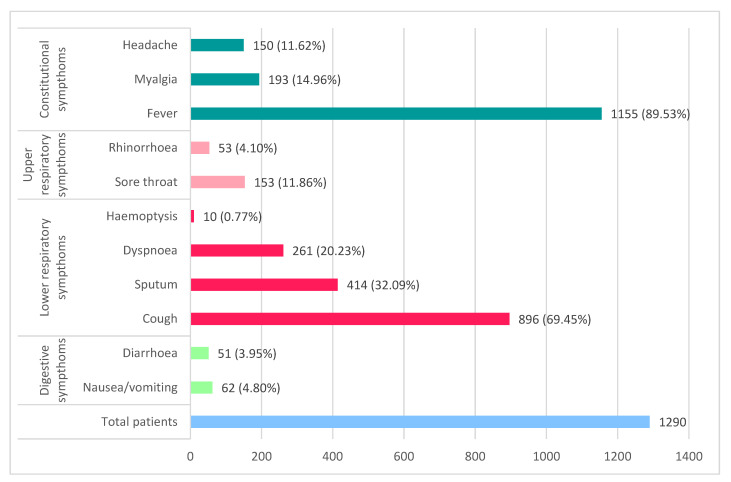
Symptomatology of COVID-19 (based on data published by Guan et al. [13] and Zhou et al. [11]).

**Figure 2 diagnostics-12-02823-f002:**
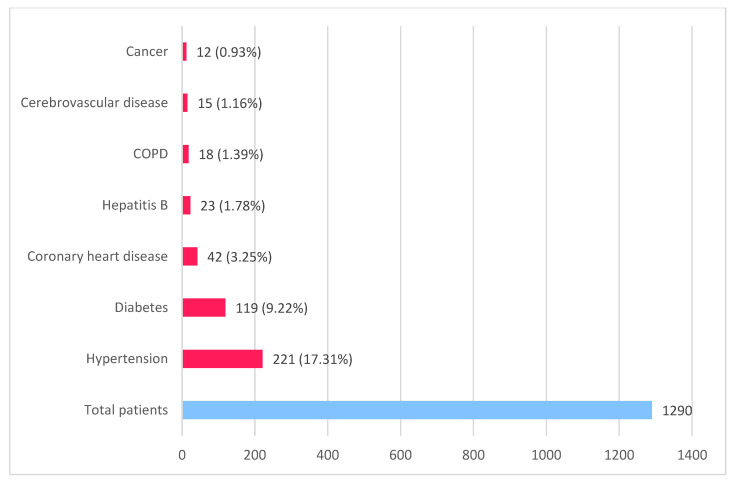
Main comorbidities in COVID-19 patients (based on data published by Guan et al. [13] and Zhou et al. [11]).

**Figure 3 diagnostics-12-02823-f003:**
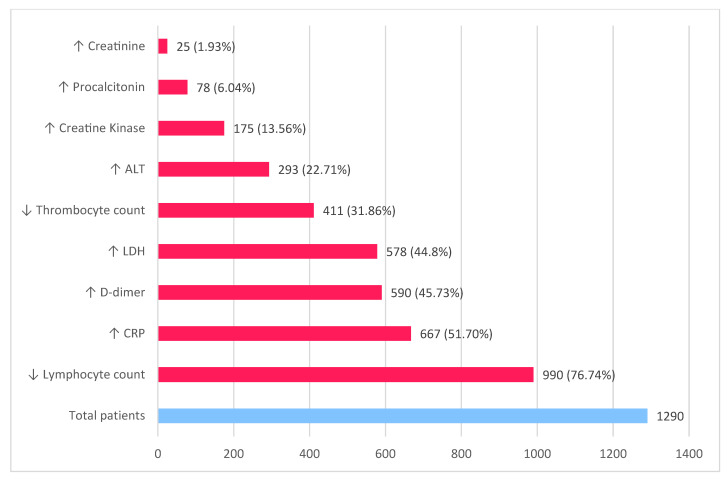
Laboratory findings in COVID-19 patients (based on data published by Guan et al. [13] and Zhou et al. [11]).

**Figure 4 diagnostics-12-02823-f004:**
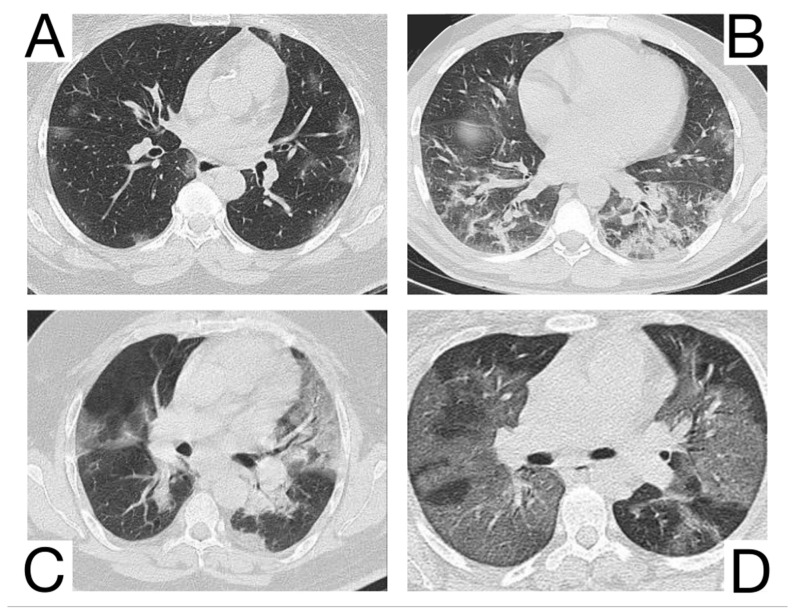
Imagistic findings in COVID-19 (**A**). Multiple areas of ground glass infiltration (patient on the third day of symptoms) (**B**). Bilateral patches of ground glass and subsegmental consolidation (**C**). Ground glass and consolidation with air bronchogram (8 days after onset) (**D**). Diffuse ground glass infiltration (white lung appearance). Note. Adapted from Hefeda et al. (2020) [22].

**Figure 5 diagnostics-12-02823-f005:**
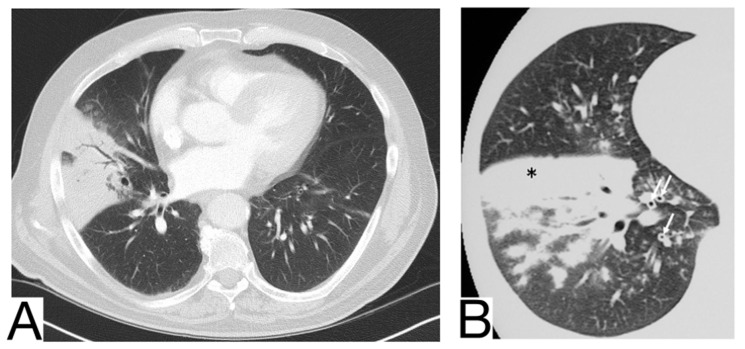
(**A**). Consolidation with air bronchogram in a patient with bacterial pneumonia (**B**). Consolidation and bronchovascular bundle thickening in a patient with Mycoplasma pneumoniae pneumonia. Note. Adapted from Mikael Häggström, M.D (**A**) and Tanaka (2016) (**B**).

**Figure 6 diagnostics-12-02823-f006:**
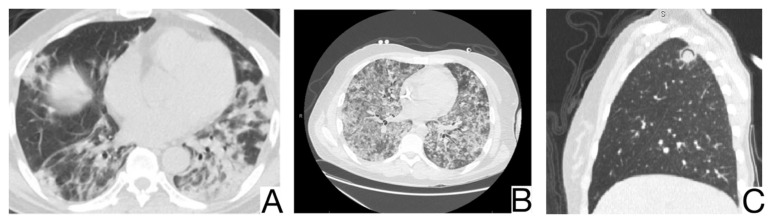
(**A**). Multifocal areas of poorly defined focal consolidation in a patient positive for influenza A (H1N1) (**B**). Bilateral ground-glass opacities and dense airspace consolidations in a patient with *Pneumocystis jirovecii* pneumonia (**C**). Halo sign in a patient with aspergillosis. Note. Adapted from Elmokadem et al. (2021) [40] (**A**), Sullivan et al. [41] (2020) (**B**), and Dr. Laughlin Dawes (**C**).

**Figure 7 diagnostics-12-02823-f007:**
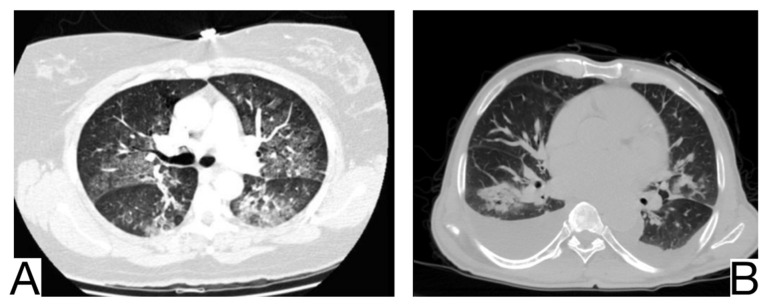
Acute pulmonary edema (**A**). Ground glass opacity in mainly perihilar and dependent distribution (**B**). Bilateral airspace opacification in central peribronchovascular distribution and smooth interlobular septal thickening (indicating interstitial edema) and moderate bilateral pleural effusion. Note. Case courtesy of The Radswiki, Radiopaedia.org, rID: 1183 (**A**) and Dr. Rania Adel Anan, Radiopaedia.org, rID: 95825 (**B**).

**Figure 8 diagnostics-12-02823-f008:**
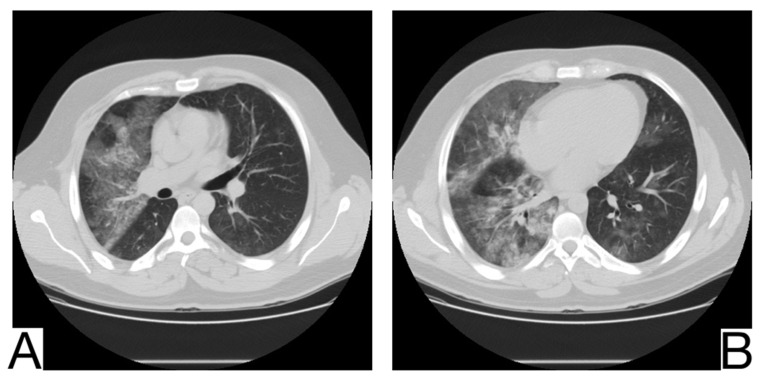
Diffuse alveolar hemorrhage, pulmonary parenchyma with a diffuse increase in density, and bilateral alveolar filling pattern, predominantly in lower lobes (**A**,**B**). Note—case courtesy of Dr. Jesus Sanchez Castro, Radiopaedia.org, rID: 68769.

**Figure 9 diagnostics-12-02823-f009:**
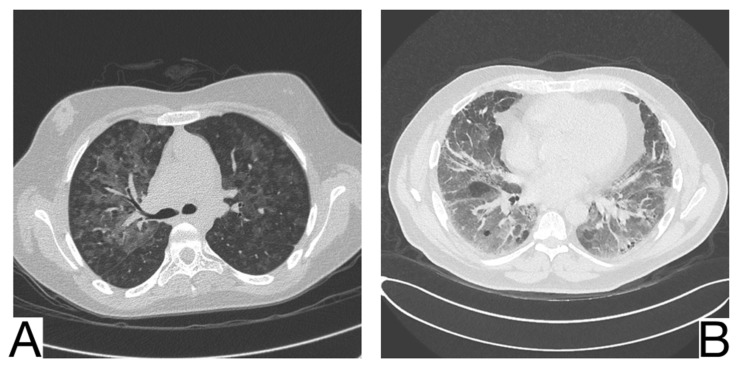
Hypersensitivity pneumonia (**A**). Perihilar ground glass changes (**B**). Gas trapping on the expiratory image, ground glass opacity, and honeycombing. Note. Case courtesy of Dr. Yi-Jin Kuok, Radiopaedia.org, rID: 17192 (**A**) and Dr. Henry Knipe, Radiopaedia.org, rID: 48107 (**B**).

**Figure 10 diagnostics-12-02823-f010:**
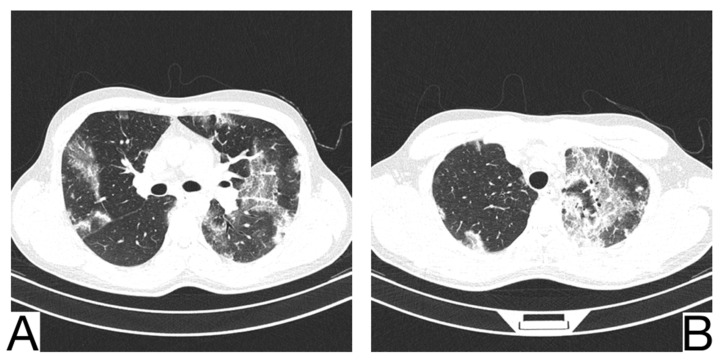
Chronic eosinophilic pneumonia (**A**,**B**). Consolidation throughout both lungs. Note. Case courtesy of Dr. Henry Knipe, Radiopaedia.org, rID: 39331.

**Figure 11 diagnostics-12-02823-f011:**
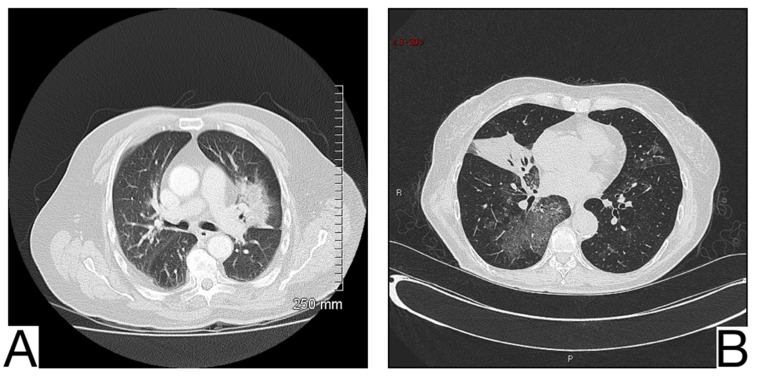
Lipoid pneumonia (**A**). Ill-defined airspace opacity next to the left hilum (**B**). Atelectasis of middle lobe and ground-glass opacity. Note. Case courtesy of Dr. Aneta Kecler-Pietrzyk, Radiopaedia.org, rID: 62113 (**A**) and Dr. Abraão Kupske, Radiopaedia.org, rID: 55752 (**B**).

**Figure 12 diagnostics-12-02823-f012:**
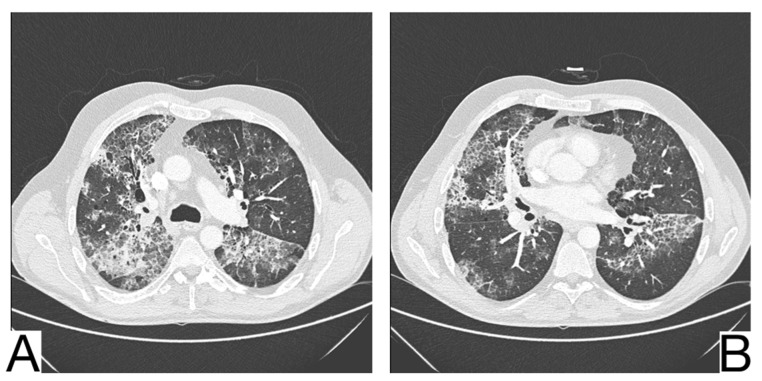
Pulmonary alveolar proteinosis (**A**,**B**). Bilateral areas of crazy paving. Note. Case courtesy of Dr. Adrià Roset Altadill, Radiopaedia.org, rID: 74896.

**Figure 13 diagnostics-12-02823-f013:**
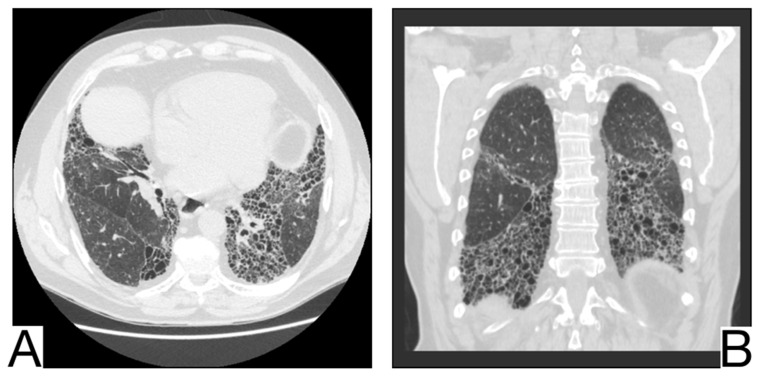
Idiopathic interstitial pneumonia (**A**,**B**). Bilateral septal thickening and honeycombing with more severe involvement toward the lung bases. Note. Case courtesy of Dr. Hani Makky Al Salam, Radiopaedia.org, rID: 41974.

**Figure 14 diagnostics-12-02823-f014:**
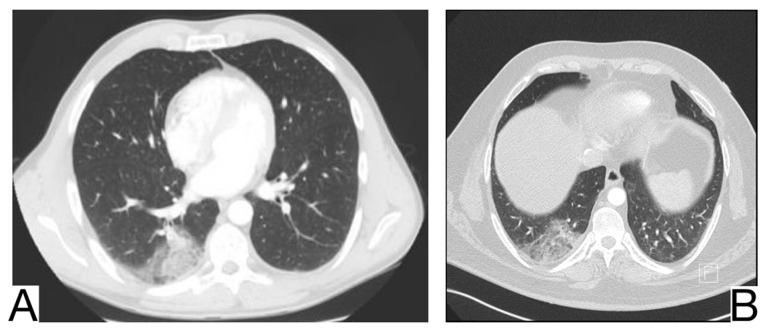
Pulmonary infarction (**A**,**B**). Wedge-shaped peripheral consolidation, absent air bronchogram. Note. Case courtesy of Dr. Vladislav Boyanov Rusinov, Radiopaedia.org, rID: 19479 (**A**) and Assoc Prof Craig Hacking, Radiopaedia.org, rID: 73062 (**B**).

## Data Availability

Not applicable.

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
