# Peer review of "Challenges in the Differential Diagnosis of COVID-19 Pneumonia: A Pictorial Review"

_diagnostics, 2022, doi:10.3390/diagnostics12112823_

Round 1

Reviewer 1 Report

A great problem is an early and repeatable imaging diagnosis. Important for hospitalisation, ICU, emergency in dyspneaThe imaging procedures are lacking, especially lung ultrasound published in the last two years, first from China and world wide in a lot of studies. Lung Ultrasound is a better stethoskop. Than you see what you mean to hear.

Author Response

Dear reviewer,

Thank you very much for your review and constructive suggestions. 

We improved our manuscript by adding a section highlighting the most important aspects of lung ultrasound.

We have extensively edited the paper (both the information and the page display of the text) to improve its quality and make it easier to follow. We have rewritten some paragraphs to convey the information more clearly and eliminated repeating information.

Reviewer 2 Report

I read with great attention the article by Marginean et al. with the title: 

 Challenges in the Differential Diagnosis of COVID-19 2 Pneumonia.

In my opinion, the article is a pictorial review and this should be highlighted in the title so the reader can expect what going on.

Some passages are repetitive and the image-only Ct scan deserves an arrow to explain to the non-radiologist their finding.

The review does not consider lung ultrasound. 

In summary, the flow of the article should be explained to the reader. Some message is redundant and the image needs to be explained. The English should be improved to a superior level. The reference is too much but some can be changed with the following lectures:

1) Zanella A, Florio G, Antonelli M, Bellani G, Berselli A, Bove T, Cabrini L, Carlesso E, Castelli GP, Cecconi M, Citerio G, Coloretti I, Corti D, Dalla Corte F, De Robertis E, Foti G, Fumagalli R, Girardis M, Giudici R, Guiotto L, Langer T, Mirabella L, Pasero D, Protti A, Ranieri MV, Rona R, Scudeller L, Severgnini P, Spadaro S, Stocchetti N, Viganò M, Pesenti A, Grasselli G; COVID-19 Italian ICU Network. Time course of risk factors associated with mortality of 1260 critically ill patients with COVID-19 admitted to 24 Italian intensive care units. Intensive Care Med. 2021 Sep;47(9):995-1008. doi: 10.1007/s00134-021-06495-y. Epub 2021 Aug 9. PMID: 34373952; PMCID: PMC8351771.

2) Venturini S, Orso D, Cugini F, Crapis M, Fossati S, Callegari A, Pellis T, Tonizzo M, Grembiale A, Rosso A, Tamburrini M, D'Andrea N, Vetrugno L, Bove T. Classification and analysis of outcome predictors in non-critically ill COVID-19 patients. Intern Med J. 2021 Apr;51(4):506-514. doi: 10.1111/imj.15140. Epub 2021 Apr 9. PMID: 33835685; PMCID: PMC8250466.

3) Vetrugno L, Baciarello M, Bignami E, Bonetti A, Saturno F, Orso D, Girometti R, Cereser L, Bove T. The "pandemic" increase in lung ultrasound use in response to Covid-19: can we complement computed tomography findings? A narrative review. Ultrasound J. 2020 Aug 17;12(1):39. doi: 10.1186/s13089-020-00185-4. PMID: 32785855; PMCID: PMC7422672.

I will be available to read your next version.

Best Regards

Author Response

Dear reviewer,

Thank you very much for your review and constructive suggestions. We have extensively edited the paper (both the information and the page display of the text) to improve its quality and make it easier to follow. We have rewritten some paragraphs to convey the information more clearly and eliminated repeating information.

Q1:In my opinion, the article is a pictorial review and this should be highlighted in the title so the reader can expect what going on.

A1: Thank you for pointing this out. We modified the title accordingly.

Q2:Some passages are repetitive, and the image-only Ct scan deserves an arrow to explain to the non-radiologist their finding.

A2: We revised the whole manuscript and corrected the repetitive phrases.

Q3:The review does not consider lung ultrasound.

A3: We added a paragraph in which we highlighted the essential aspects of ultrasound importance.

Q4:In summary, the flow of the article should be explained to the reader. Some message is redundant and the image needs to be explained. The English should be improved to a superior level. The reference is too much but some can be changed with the following lectures:

1) Zanella A, Florio G, Antonelli M, Bellani G, Berselli A, Bove T, Cabrini L, Carlesso E, Castelli GP, Cecconi M, Citerio G, Coloretti I, Corti D, Dalla Corte F, De Robertis E, Foti G, Fumagalli R, Girardis M, Giudici R, Guiotto L, Langer T, Mirabella L, Pasero D, Protti A, Ranieri MV, Rona R, Scudeller L, Severgnini P, Spadaro S, Stocchetti N, Viganò M, Pesenti A, Grasselli G; COVID-19 Italian ICU Network. Time course of risk factors associated with mortality of 1260 critically ill patients with COVID-19 admitted to 24 Italian intensive care units. Intensive Care Med. 2021 Sep;47(9):995-1008. doi: 10.1007/s00134-021-06495-y. Epub 2021 Aug 9. PMID: 34373952; PMCID: PMC8351771.

2) Venturini S, Orso D, Cugini F, Crapis M, Fossati S, Callegari A, Pellis T, Tonizzo M, Grembiale A, Rosso A, Tamburrini M, D'Andrea N, Vetrugno L, Bove T. Classification and analysis of outcome predictors in non-critically ill COVID-19 patients. Intern Med J. 2021 Apr;51(4):506-514. doi: 10.1111/imj.15140. Epub 2021 Apr 9. PMID: 33835685; PMCID: PMC8250466.

3) Vetrugno L, Baciarello M, Bignami E, Bonetti A, Saturno F, Orso D, Girometti R, Cereser L, Bove T. The "pandemic" increase in lung ultrasound use in response to Covid-19: can we complement computed tomography findings? A narrative review. Ultrasound J. 2020 Aug 17;12(1):39. doi: 10.1186/s13089-020-00185-4. PMID: 32785855; PMCID: PMC7422672.

 A4: We have revised the errors and improved the references with those suggested.

Round 2

Reviewer 1 Report

Congratulations

Reviewer 2 Report

Thank you for addressing my comments.

Please revise the English to do to the paper a better fluency.

Best Regards